# Quantitative Proteomics Reveals Molecular Network Driving Stromal Cell Differentiation: Implications for Corneal Wound Healing

**DOI:** 10.3390/ijms23052572

**Published:** 2022-02-25

**Authors:** Krishnatej Nishtala, Trailokyanath Panigrahi, Rohit Shetty, Dhanananajay Kumar, Pooja Khamar, Rajiv R. Mohan, Vrushali Deshpande, Arkasubhra Ghosh

**Affiliations:** 1GROW Research Laboratory, Narayana Nethralaya Foundation, Bangalore 560099, India; tejnishtala@gmail.com (K.N.); trailoknath@narayananethralaya.com (T.P.); dhanurai17@gmail.com (D.K.); 2Cornea and Refractive Services, Narayana Nethralaya, Bangalore 560010, India; drrohitshetty@yahoo.com (R.S.); dr.poojakhamar@gmail.com (P.K.); 3Harry S. Truman Memorial Veterans’ Hospital, Columbia, MO 65201, USA; mohanr@missouri.edu; 4Department of Veterinary Medicine & Surgery and Biomedical Sciences, College of Veterinary Medicine, University of Missouri, Columbia, MO 65212, USA; 5College of Veterinary Medicine, University of Missouri, Columbia, MO 65211, USA; 6Mason Eye Institute, School of Medicine, University of Missouri, Columbia, MO 65201, USA

**Keywords:** cornea, wound healing, TGF-β, keratocytes, fibroblasts, myofibroblasts, quantitative proteomics, LCMS, SLIT-ROBO signaling, integrin signaling, actin cytoskeleton, calcium signaling

## Abstract

The differentiation of keratocytes to fibroblasts and myofibroblasts is an essential requisite during corneal wound closure. The aim of this study is to uncover factors involved in differentiation-dependent alteration in the protein profile of human corneal stromal cells using quantitative proteomics. Human corneal fibroblasts were cultured and differentiated into keratocytes in serum-free media and myofibroblasts through treatment with TGF-β. The protein cell lysates from the donors were tryptic and were digested and labeled using a 3-plex iTRAQ kit. The labeled peptides were subjected to LCMS analysis. Biological functional analysis revealed a set of crucial proteins involved in the differentiation of human corneal stromal cells which were found to be significantly enriched. The selected proteins were further validated by immunohistochemistry. Quantitative proteomics identified key differentially expressed proteins which are involved in cellular signaling pathways. Proteins involved in integrin signaling (Ras-RAP1b, TLN and FN) and SLIT-ROBO pathways (PFN1, CAPR1, PSMA5) as well as extracellular matrix proteins (SERPINH1, SPARC, ITGβ1, CRTAP) showed enhanced expression in corneal fibroblasts and myofibroblasts compared to keratocytes, indicating their possible role in wound healing. Corneal stromal cell differentiation is associated with the activation of diverse molecular pathways critical for the repair of fibroblasts and myofibroblasts. Identified proteins such as profilin 1 and talin could play a tentative role in corneal healing and serve as a potential target to treat corneal fibrosis.

## 1. Introduction

Visual clarity depends on a clear and transparent cornea. Trauma to the cornea due to corrective refractory surgical procedures such as photorefractive keratectomy (PRK) and mechanical or chemical injuries such as corneal tear, abrasion, burn, etc., can cause an aberrant corneal stromal wound repair response leading to haze or opacity that can distort vision [1,2]. A typical wound healing response involves activation of quiescent stromal keratocytes (KCT) differentiating into fibroblasts (HCF) and myofibroblasts (MYO) [3]. Myofibroblasts are fundamental to the wound healing process; however, their excessive or prolonged activation leads to scarring [4,5], typically characterized by an excess of collagen deposition and α-smooth muscle actin (α-SMA) expression [6,7]. Therefore, molecular signaling and secretory function of myofibroblasts is an essential regulatory component in the wound repair process. The important cellular interaction in corneal healing involves the recruitment of inflammatory cells into the corneal stroma post−epithelial injury [8,9]. The activation and differentiation of keratocytes has been shown to occur as a response to the stimuli by interleukins 1α/1β (IL-1α/IL-1β) released by corneal epithelial cells as a consequence of injury [2,10]. In the case of proper wound healing, the differentiated fibroblasts and myofibroblasts undergo apoptosis upon wound closure, restoring corneal transparency [10]. However, in certain scenarios (e.g., flap detachment or post-LASIK ectasia), aberrant or uncontrolled wound healing leads to fibrosis (haze or scarring) with reduced corneal transparency or increased corneal opacity [11]. Hence, it is pertinent to understand the molecular mechanisms that modulate corneal wound healing at the early stages when stromal keratocytes are activated and undergo the differentiation to the repair cell types. Identification of such molecular factors could provide the putative targets to modulate the wound healing process and reduce the incidence of corneal fibrosis in the clinical scenario.

Previous studies have identified some of the factors that stimulate the activation and differentiation of keratocytes to fibroblasts and myofibroblasts. TGF-β and PDGF have been shown to be powerful differentiation and chemotactic agents for fibroblasts which can initiate wound contraction [5,12]. TGF-β has been shown to drive corneal wound healing, where TGF-β isoforms regulate myofibroblast differentiation [13]. In vivo studies have shown the TGF-β superfamily is structurally related [14,15] where these multifunctional growth factors play prominent roles in fibrosis via myofibroblast differentiation [4,5,16,17,18], embryonic development, granulation [14], cell contraction and loss of transparency, specifically in corneal tissue. TGF-β can maintain its production in an autocrine manner in myofibroblasts, necessitating additional mechanisms to halt the stromal repair processes once the wound is closed [9]. Mulhollend et al. [19] demonstrated the molecular interplay of MMP-2 and -9 in wound healing in a rabbit eye model. The study suggests MMP-9 as a candidate factor in stromal repair. In studies on knock-out mice, MMP-9 has been shown to increase re-epithelization and has been shown to be associated with basement membrane formation [19]. MMP-2, -9 and -3 have all been connected to keratocyte migration and fibroblast activation at the wound site, in addition to the construction of the basement membrane and stromal remodeling [19]. Recently, it has been shown that epithelium-derived factors may predispose certain subjects to develop corneal haze post−refractive surgery through PREX1, which also regulates the TGF-β pathway [20]. Therefore, regulating TGF-β-dependent pathways for modulation of wound healing and preventing visual loss due to scarring is attractive.

Though topical use of mitomycin C (MMC) post-operatively has been the practice to prevent fibrosis and corneal haze, its effects on stromal integrity have been shown to be debatable [21], at least in cases with low-diopter corrections; hence, a targeted approach to prevent fibrosis is the need of the hour. Although transforming growth factor-beta (TGF-β)-dependent wound healing and myofibroblast differentiation models have been well documented in the literature, a global analysis of the protein profiles of such cells undergoing a differentiation process has not been reported. Therefore, in the current study, using quantitative proteomics, we aim to unravel the complete proteome profiles of keratocytes, differentiated fibroblasts and myofibroblasts to identify molecular processes and factors which may be targeted for treating corneal haze.

## 2. Results

### 2.1. Determination of Differentiation State of Corneal Stromal Cells

The corneal stromal cell phenotypes—keratocytes, fibroblasts and myofibroblasts—were analyzed morphologically for their differentiation (Figure 1I (D–F)). The differentiation states of the three cell phenotypes were validated using cell-type-specific markers by quantitative PCR analysis. The expression of gene Aldh3a1 [22] was higher in keratocytes compared to fibroblasts and myofibroblasts (Figure 1I (A)) as described earlier [23]. Similarly, Thy-1 [23] (Figure 1I (B)) and αSMA [24] (Figure 1I (C)) were observed to be expressed higher in fibroblasts and myofibroblasts compared to keratocytes, indicating their respective differentiated states in vitro. The enhanced expression of each cell-specific gene is an indication of the differentiation into the respective cell type. The differentiation states of these stromal cells were also confirmed by fluorescence microscopy (Figure 1II) and Western blotting (Appendix A). The expression of Thy1 was higher in fibroblasts as compared to keratocytes and myofibroblasts. The keratocyte (ALDH3A) and myofibroblast (αSMA) markers also showed higher expression in the respective cell types, which confirmed successful differentiation of fibroblasts in vitro (Figure 1II).

### 2.2. Quantitative Proteomics Reveals Enhanced Cytoskeletal Regulation and Cell Signaling

Equal quantities of whole-cell lysates of keratocytes, fibroblasts and myofibroblasts separated using SDS-PAGE showed distinct protein expression patterns (Figure 1III (A)). To investigate changes in the proteome during stromal cell differentiation, quantitative proteomics experiments using iTRAQ labeling were performed in four biological replicates (Figure 1III (B)) consisting of cells derived from four different donors. A total ion chromatogram of KCT, HCF and MYO showed <10% of variation in the intensities, indicating equal protein loading (Appendix A). A total of two hundred and seventy proteins were identified with ≥2 peptides and ≤1% false discovery rate (FDR) which were common in all the biological replicates and were efficiently labeled (Appendix A). Panel A in Figure 2 represents the volcano plot of all biological replicates, highlighting the most important proteins in fibroblasts and myofibroblasts which illustrate the statistical significance (log p value) vs. fold-change of differentially expressed proteins. Panel B in Figure 2 represents a heatmap of the differentially expressed 105 proteins across fibroblasts and myofibroblasts when compared to keratocytes in all replicates with a protein coverage of >30%. We further analyzed the proteins for known functions by Panther [25] and represent the different protein classes identified as a proportion of the total in panel C (Figure 2). The data show that proteins with nucleic acid binding (26.3%), cytoskeletal proteins (12.3%) and calcium-binding proteins (3.9%) were most abundant, whereas transmembrane receptor regulatory protein (0.4%) and cell junction protein (0.4%) were low in abundance (panel C, Figure 2). Reactome pathway analysis of the altered proteins revealed axon guidance (*p* = 2.95 × 10^−14^), nonsense-mediated decay (*p* = 3.94 × 10^−14^), signaling by ROBO receptors (*p* = 2.16 × 10^−12^) and regulation of expression of SLITs and ROBOs (and integrin signaling pathways as highly altered (*p* = 4.02 × 10^−4^) (Table 1).

Increased levels of the integrin signaling proteins fibronectin (FN), talin 1 (TLN1) and Ras-related protein (RAPB1) were observed, complementing the actin cytoskeletal regulation (Figure 3I (a–d)). The extracellular matrix proteins collagen 1A1 (COL1A1), integrin subunit beta 1 (ITGB1), osteonectin (OSN), cathepsin B (CTSB) and serpin family H member 1 (SERPINH1) were identified at higher levels in fibroblasts and myofibroblasts, indicating active cytoskeletal reorganization during the differentiation from keratocytes (Figure 3II (a–f)). Elevated cofilin (CFN), calponin 2 (CNN2), moesin (MSN) and annexin A2 (ANXA2), observed in both fibroblasts and myofibroblasts, play significant roles in the remodeling of extracellular matrix and actin cytoskeletal regulation (Figure 3III (a–f)). Proteins involved in signaling by ROBO receptors such as profilin 1 (PFN1), caprin-1 (CAPR1) and lamin B1 (LMNB1) showed altered expression (Figure 3IV (a–d)). These results indicate that the differentiation of dormant keratocytes into active fibroblasts and myofibroblasts involves distinct molecular pathways with enhanced expression of extracellular matrix and actin cytoskeletal proteins alongside altered integrin and SLIT-ROBO pathway signaling (Figure 3I–IV). Moreover, pyruvate kinase (PKM), phosphoglycerate kinase 1 (PGK1), alpha-enolase (ENOA), hexokinase (HK1) and malate dehydrogenase (MDH2) proteins were altered, which are involved in glucose metabolism.

### 2.3. Pathway—Protein Interaction Networks

The pathway—protein interaction network was constructed and loaded onto Cytoscape V 3.6.1. The network was further analyzed based on the edge-betweenness centrality. Edge-betweenness centrality is defined as the frequency of an edge that is placed on the shortest paths between all pairs of vertices. The edges with highest betweenness values are most likely to lie between subgraphs. Nodes representing pathways and edges correspond to the interactions of proteins and pathways. The size of the nodes was decided by edge-betweenness centrality. Furthermore, we calculated the edge-betweenness score to find the most important protein—pathway interactions in the co-expression network. As a result (refer to the details on edge-betweenness of co-expression networks in Appendix A), the interaction between glucose metabolism and the metabolism of proteins had the highest number of shortest paths, indicating a change in the metabolism and energy production as the cells undergo differentiation from keratocytes to the repair phenotypes. The interactome is shown in Figure 4A. A list of the top 29 pathway—protein names on the basis of betweenness can be found in Appendix A. The proteins involved in the highest-degree-associated pathways are represented in the form of heatmaps across all biological replicates in fibroblasts and myofibroblasts (Figure 4B–F) normalized to their respective biological keratocyte data. The highest degrees were found to be assigned to the metabolism of proteins, extracellular matrix organization, axon guidance, glucose metabolism and SLIT-ROBO signaling. GAB1–GRB2-associated-binding protein 1, involved in intracellular signaling cascades, showed the highest degree of edge-betweenness. We note that a variety of ribosomal proteins were identified across the different pathways, likely due to both fibroblasts and myofibroblasts being metabolically active with higher levels of translational activity compared to quiescent keratocytes. This analysis identifies a number of novel proteins and pathways involved in human stromal cell differentiation.

### 2.4. Validation by Immunofluorescence, Western Blot and Immunohistochemistry

Validation of the most significantly altered proteins in each of the altered pathways was performed by Western blotting and quantified from one independent biological replicate. Quantification of densitometry results demonstrate cofilin (COF) (Figure 5A (ii)), profilin 1 (PROF1) (Figure 5A (iii)) and annexin A2 (ANXA2) (Figure 5A (iv)) to be elevated in fibroblasts and myofibroblasts, validating the observations in quantitative proteomics. ANXA2 (P07355) was detected as two bands on Western blotting (Figure 5A (i)), indicating two isoforms: -isoform 1 (P07355-1: 38.6 KDa) and isoform 2 (P07355-2: 40.4 KDa). The Western blot protein band density values (derived from ImageJ software) were normalized to GAPDH to determine the normalized expression of the altered proteins. Since the housekeeping genes such as G3P (iTRAQ) and ACTN (data not shown) were observed to have altered during differentiation of the stromal cells, the total lane intensity of the Coomassie-stained lanes (10 µg lysate for each cell type) has been used for normalization. Selected proteins that showed differential expression in keratocytes, fibroblasts and myofibroblasts in the proteomics experiments were further validated in fibrotic tissue derived from patients with corneal scars. We investigated corneal tissues from three different kinds of corneal scar subjects—viral keratitis, fungal keratitis and corneal ulcerative scar—keeping clear corneas from healthy donors as control. profilin 1, cofilin, filamin A and cathepsin B expression varied in fibrotic tissue as compared to the healthy, donor controls. (Figure 5B (I–IV)). Profilin 1 and cofilin expression was particularly elevated in the case of viral keratitis and fungal keratitis compared to corneal ulcers. Cathepsin B expression also increased in all three fibrotic conditions, particularly in the stromal region (Figure 5B (IV)). The increased expression of these actin cytoskeletal proteins in the fibrotic cornea suggests myofibroblastic activation of stromal fibroblasts upon infection.

Therefore, irrespective of the cause of fibrosis in the corneal stroma, these key proteins are elevated, thus indicating the presence of myofibroblasts that produce a disordered extracellular matrix, driving visual compromise.

## 3. Discussion

Corneal wound healing involves the differentiation of quiescent keratocytes to active fibroblasts and myofibroblasts which proliferate and secrete extracellular matrix proteins, thereby aiding wound contraction and closure [3,26,27,28]. Continued TGF-β stimulation in myofibroblasts can drive the development of fibrosis by inducing excessive collagen synthesis and ECM deposition [29,30]. To our knowledge, this is the first study to characterize changes occurring in the proteome during corneal stromal cell differentiation. In our study, the initial 1D-SDS-PAGE analysis showed that the stromal cell phenotypes differ in their proteome pattern (Figure 1III (A)), which is in agreement with a previous study [31]. Upon further investigation using quantitative proteomics, we show that the corneal stromal cell differentiation is robustly associated with enhanced extracellular matrix organization, cytoskeletal regulation, integrin signaling, altered protein metabolism, axon guidance and signaling by ROBO receptors, etc. (Figure 3I–IV). Protein network and Reactome analysis revealed alterations to a variety of signaling pathways involving key proteins with possible involvement in wound closure and stromal cell differentiation, which were previously unreported in corneal stromal cell differentiation.

The group of Ena/VASP proteins (ENAH, EVL1 and VASP) are key factors which enhance actin filament elongation [32], an important aspect of cytoskeletal reorganization during the differentiation process. The actin elongation initiates with the recruitment of profilin, making profilin-actin complexes. Our data indicate a higher level of profilin in fibroblasts and myofibroblasts compared to the keratocytes. Profilin binds to the central proline-rich domain of an Ena/VASP protein [32] and has been demonstrated to play a critical role in actin cytoskeleton elongation [33]. Since profilin expression was higher in the human donor corneas with scars, it appears to be an important factor in aberrant corneal stromal reorganization and the scarring process.

Proteins such as SLITs have been shown to be associated with cell membranes and the extracellular matrix (ECM) [34,35]. The SLIT ligands bind to roundabout (ROBO) transmembrane receptors, which regulate axonal guidance, cell migration and differentiation of keratocytes to fibroblast and myofibroblasts [35]. The elevated ROBO levels observed in our experiments could therefore be important for both the stromal healing process as well as corneal nerve regeneration since during surgery or corneal trauma, the corneal nerves are also expected to be affected.

Cellular homeostasis is achieved or maintained by specific signaling cascades, which are important in maintaining the actin cytoskeletal structure and integrity of the extracellular matrix structure [36] (Appendix A). The integrin signaling pathway is activated in response to mechanical stimuli from the extracellular matrix, leading to intracellular events. The α and β subunits of the integrin complexes dissociate upon activation [37]. Talin, which is the key regulator of integrin activation, was found to be 1.2-fold increased in the differentiated myofibroblasts and fibroblasts compared to keratocytes (Appendix A). Talin binds to the β subunit of integrins, which dissociates the two integrin subunits after activation. Integrin signal activation by talin plays an important regulatory role in linking the extracellular matrix to cytoskeleton reorganization [37,38]. Keratocytes secrete high levels of collagen and proteoglycan [39,40,41,42]. Stromal collagen alpha-1(VI) chain (COL6A1) was identified at reduced levels in myofibroblasts and fibroblasts as compared to keratocytes but was not statistically significant. However, levels of collagen alpha-1(I) chain (COL1A1), collagen alpha-2(I) chain (COL1A2) and collagen alpha-3(VI) chain were not significantly altered, although it is possible that in monolayer cultures, significant amounts of collagen deposition may take a longer time, similar to the in vivo process which usually takes several weeks post-insult to develop.

One of the important proteins of the extracellular matrix is fibronectin (FN), which is present in the form of fibrils connecting the cells to the ECM. The expression of fibronectin, expectedly, was found to be higher in myofibroblasts as compared to keratocytes and fibroblasts. The extracellular matrix proteins SERPINH1 (HSP47), SPARC (osteonectin), and cartilage-associated protein (CRTAP) were expressed at higher levels in myofibroblasts and fibroblasts as compared to keratocytes.

HSP47 is a 47 KDa collagen-binding protein that binds to triple helix collagen [43]. It has been reported that Hsp47 knockout fibroblast cells, due to the delay in secreting collagen, aggregate in the form of procollagen and accumulate in ER and are degraded by autophagy mechanisms. This results in the formation of abnormal, thin and branched collagen fibrils [44]. The expression of Hsp47 was found to be higher in fibroblasts and myofibroblasts as compared to keratocytes. SPARC is another protein that plays an important role in tissue remodeling and cell–ECM interactions [45]. Cartilage-associated protein (CRTAP) was also found to be upregulated in myofibroblasts and fibroblasts. CRTAP has been recognized in performing post-translational modifications, extracellular fibril assembly and intracellular trafficking. Delay in the collagen helix folding and excess modification by lysyl hydroxylase and prolyl 4-hydroxylase is due to lack of CRTAP function [46]. Besides the extracellular matrix, proteins involved in cytoskeletal organization and ROBO receptor signaling showed increased expression of fibroblasts and myofibroblasts as compared to keratocytes. Cofilin (COF), annexin 2 (ANXA2), moesin (MSN) and actin cytoskeletal proteins cofilin (COF), profilin 1 (PROF1) and myosin 9 (MYH9) were observed at elevated levels in fibroblasts and myofibroblasts compared to keratocytes, indicating active cytoskeletal reorganization and Rho GTPase signaling. Both COF and PROF1 are known to activate actin filament polymerization essential for cytoskeletal reorganization. Elevated levels of COF, PROF1, tropomyosin II and Rho GDP dissociation inhibitor were observed in human fetal lung fibroblast cell lines (HFL-1) treated 24 h with TGF-β by two-dimensional gel proteomic analysis [47], but their alteration in corneal conditions was not known previously. Though we could not observe significant changes in Rho GTPase protein levels, the Rho GDP dissociation inhibitor tropomyosin II (TPM2) was identified at elevated levels in corneal fibroblasts and myofibroblasts, indicating increased activity of the cellular contractile machinery. COF1 and ANXA2 signaling is also important for the regulation of inflammatory signaling via lysosomal and inflammasome activation pathways [48]. Therefore, participation of the differentiated stromal cells in the regulation of inflammation is possibly important for controlled wound healing.

Most of the identified proteins were involved in processes such as apoptosis, cytoskeleton organization, protection against stress, fibrosis, etc. While similar processes were observed in the proteomic analysis of the soluble fraction from human corneal fibroblasts reported by Karring et al. [31], the comparative analysis with quiescent keratocytes and myofibroblasts provided a pathologic context and helped reveal many more novel entities.

A limitation of the current study is that the in vitro primary stromal cell differentiation model, although a close approximation of the in vivo stromal cell differentiation process, is not an exact physiological mimic. The tissue-specific cues from an injured natural stroma may be more appropriately studied in an in vitro 3-D model for fibrosis using collagen matrices. However, to address this, we validated few of these markers in human donor corneas with fibrosis. We observed higher expressions of cathepsin B, cofilin and profilin 1 in fibrotic tissue, which validated our observations from the in vitro model. The recently reported proteomic profile of human corneal stroma by Khamar et al. [20] has an overlap of 111 proteins with the in vitro primary stromal cell differentiation model (Figure 6A). The list of 111 proteins is shown in Figure 6B. A few important cytoskeletal proteins such as collagens (COL6A1, COL1A2), actinin 4 (ACTN4) and 14-3-3 class of proteins, which are known to be involved in early wound healing responses, were also found in this stromal cell differentiation study (Appendix A).

Taken together, quantitative proteomic profiling provides a comprehensive dataset of corneal stromal cells undergoing differentiation to the reparative cell phenotypes and identifies important molecular changes and signaling pathways that likely regulate corneal stromal cell differentiation (Figure 7). A greater understanding of proteins such as ANXA2, cathepsin B, cofilin, profilin 1, etc., in corneal wound healing can possibly help modulate fibrosis in a targeted fashion with minimal or no damage to the corneal integrity, unlike the current modes of treatment such as mitomycin C (MMC).

## 4. Materials and Methods

### 4.1. Cell Culture and Differentiation of Human Corneal Fibroblasts

In vitro culture and differentiation of human corneal fibroblasts to myofibroblasts and keratocytes was carried out as reported earlier [49,50,51]. Corneal buttons were obtained from four donor cadaver eyes without any clinical history of ocular disease and free of any infections. The donor eye tissues were obtained from the institute eye banks at Narayana Nethralaya. The eyes that were not used for surgical transplants were donated to the laboratory to be used in the experiments. The corneal button was washed with sterile 1× phosphate-buffered saline containing 1× antibiotics solution. The epithelium and endothelium were removed with gentle scraping using a sterile surgical blade, and the corneal stroma was chopped into small pieces and cultured in Dulbecco’s modified eagle medium F12 (DMEM-F12) supplemented with 10% fetal bovine serum (FBS) (Gibco, ThermoFisher Scientific, Waltham, MA, USA) and 1% penicillin, streptomycin and actinomycin for 36 h to achieve confluency. The primary fibroblast cell lines obtained from each of the four donor corneas were not immortalized and used for the experiments described. Confluent fibroblasts (approximately 4–5 million cells) were differentiated in vitro into keratocytes under serum-free conditions for 96 h with media replenished every 36 h. For differentiation into myofibroblasts, approximately 60,000–70,000 fibroblast cells were treated with TGF-β at a concentration of 1 ng/mL every 24 h for 5 d or 120 h. HMK1 (BR4) primary human corneal fibroblast cell line was obtained from Dr. Rajiv R Mohan.

### 4.2. Gene Expression and Quantitative PCR (qPCR) Analysis

RNA isolation and cDNA synthesis from corneal stromal cells was performed as described earlier [10]. Briefly, total ribonucleic acid (RNA) from the harvested cells was isolated using TRIzol reagent (Thermo Fisher, Carlsbad, CA, USA), and the RNA concentration was estimated using Eppendorf BioPhotometer Plus (Eppendorf, Hamburg, Germany). Complementary DNA (cDNA) from RNA was synthesized using Bio-Rad iScript™ cDNA synthesis kit (Bio-Rad, Hercules, CA, USA) as per the manufacturer’s protocol. Quantitative real-time PCR analysis for the genes of interest was performed as described in Shetty et al. [52].

### 4.3. Protein Extraction from Cultured Stromal Cells and iTRAQ Labeling

For quantitative proteomics, the harvested cell pellets dissolved in 0.5 M Triethyl Ammonium Bicarbonate (TEAB) buffer were subjected to 3 cycles of sonication on ice at 40% pulse and 40% energy using an ultrasonic homogenizer (Model 3000, Biologics Inc., Manassas, VA, USA). The cell extracts were clarified by centrifugation at 10,000 rpm for 15 min at 4 °C. The protein concentrations of the supernatants were estimated by Bradford protein assay (Bio-Rad, Hercules, CA, USA) and absorbance measured at 595 nm using bovine serum albumin as standard using Eppendorf BioPhotometer Plus (Eppendorf, Hamburg, Germany). iTRAQ labeling of equal concentrations of the protein from fibroblasts, keratocytes and myofibroblasts was performed as per the manufacturer’s instructions (SCIEX, Framingham, MA, USA). Briefly, cell lysates from each group—i.e., keratocytes, fibroblasts and myofibroblasts—were sequentially denatured, reduced, alkylated and subjected to proteolytic digestion using trypsin at an enzyme-to-protein ratio of 1:15 for 16 h. Triplex iTRAQ labeling was performed by labeling peptides from keratocytes, fibroblasts and myofibroblasts with 114, 115 and 116 labels, respectively. The pooled labeled samples were then fractionated using a cation exchange cartridge (SCIEX, Framingham, MA, USA).

Furthermore, cell lysates from three biological replicates originating from three different donors (three groups: keratocytes, fibroblasts and myofibroblasts) were also processed the same way, as discussed earlier, and were used to perform quantitative proteomic experiments.

### 4.4. Quantitative Proteomics

The iTRAQ-labeled peptides were analyzed on SCIEX Triple TOF 5600+ mass spectrometer (SCIEX, Framingham, MA, USA) coupled with Eksigent ekspert 425 nano LC. The peptides were resolved on Eksigent 3C18-CL-120, 3 µm, 120 Å, 350 µm × 0.5 mm, 0.075 µm × 150 mm analytical column coupled with Eksigent nanoLC Trap ChromXP CL 3 µm, 120 Å, 350 µm × 0.5 mm at a flow rate of 250 nL/min over a gradient of 90 min using 95% water, 5% acetonitrile and 0.1% formic acid as aqueous phase/buffer A and 95% acetonitrile, 5% water and 0.1% formic acid as organic phase/buffer B. The gradient conditions were 0–2 min at 5% buffer B, 2–4 min 10% buffer B, 4–65 min 30% buffer B, 65–70 min 90% buffer B and maintained in 90% buffer B until 78 min, 78–82 min 5% buffer B continued in 5% buffer B until 90 min. The peptides were analyzed on SCIEX 5600+ Triple TOF mass spectrometer in full-scan MS mode followed by information-dependent acquisition (IDA) mode with the top 25 intense peptides considered for fragmentation. Full-scan MS spectra were acquired in the mass range of 350–1250 *m*/*z* with an accumulation time of 200 ms, and tandem mass spectra in IDA mode were acquired in 100–1800 *m*/*z* for 50 ms with collision energy adjusted to iTRAQ reagent. To avoid redundancy, the fragmented spectra were excluded for 15 s. The resultant fragment spectra were analyzed and searched against UniProt human canonical database using Protein Pilot v 4.5 software.

### 4.5. Statistical Analysis

LCMS analysis of the iTRAQ-labeled corneal stromal cell lysate digests was carried out in triplicate. The proteins with ≥1.2-fold iTRAQ ratio and a *p*-value of < 0.05 were considered statistically significant. Functional classification of the significantly altered proteins was performed using Panther software tool [25] and Reactome pathway analysis [53].

### 4.6. Network Model Describing the Protein-Pathway Interaction

Using the information on the pathways from Reactome, the list of proteins and the pathways were loaded into Cytoscape v. 3.6.1. The output was a biological network of the number of proteins involved in a particular pathway. The analysis of the network modules of highly interacting groups of pathways was performed based on the betweenness centrality. The size of the pathway module revealed the maximum proteins involved in a particular pathway.

### 4.7. Immunofluorescence for Human Corneal Fibroblast Cells

Cells cultured in chamber slide were washed two times with cold 1× PBS after the end of treatment. Cells were fixed using 4% paraformaldehyde for 15 min at 200 m temperature, followed by two washes with 1 × PBS. Cells were permeabilized using 1% TritonX 100 for 30 min at room temperature, washed with PBS and incubated with 5% BSA in PBS for 1 h at room temperature. Cells were then incubated with primary antibody ALDH3A1 (1:200, ab76976, Abcam, Cambridge, UK), alpha SMA (1:100, ab7817, Abcam, Cambridge, UK)) and THY1 (1:200, ab133350, Abcam, Cambridge, UK) at 4 °C overnight. The cells were washed 3 times using PBS followed by Alexa Flour 488 anti-mouse (1:5000, ab150113) and Alexa Flour 647 anti-mouse (1:5000, ab150115) incubation for 1 h at room temperature. The cells were washed 3 times with PBS and mounted using FluoroShield with DAPI (F6057-20ML; Sigma, St Louis, MO, USA). Images were captured using a molecular device confocal system with 20X objective.

### 4.8. Immunohistochemistry of Fibrotic Cornea

Formalin-fixed, paraffin-embedded fibrotic and control cadaver corneas were obtained from the Narayana Nethralaya pathology department repository. IHC experiments were performed using the Dako EnVision+ System-HRP kit. Using the Leica microtome system, 4 μm thick sections were obtained, and antigen retrieval was performed using citrate buffer (pH-6) by boiling followed by washing in running water and blocking for endogenous peroxidase for 5 min at room temperature. The slides were washed with Tris-buffered saline (1 × TBS) and incubated with primary antibody filamin A (1:200, ab51217, Abcam, Cambridge, UK), profilin 1 (1:250, ab124904, Abcam, Cambridge, UK), cofilin (1:100, ab54532, Abcam, Cambridge, UK) and cathepsin B (1:100, ab58802, Abcam, Cambridge, UK) at 4 °C overnight followed by washing and incubation with Alexa Flour 488 anti-mouse (1:5000, ab150113), Alexa Flour 488 anti-rabbit (1:5000, ab150077) and Alexa Flour 647 anti-mouse (1:5000, ab150115) secondary antibodies as per the IHC kit protocol. The tissues were counterstained with hematoxylin before the substrate (DAB) reaction step. Sample slides without primary antibodies were used as negative control. Photomicrographs were taken using Olympus CKX53 (Olympus Corporation, Tokyo, Japan) with 10× objective. The protein expressions were scored as high or low by observing the intensity of the brown color staining in the tissue.

### 4.9. Immunoblotting

For validation of the differentially altered proteins by Western blotting, the following antibodies were used: rabbit monoclonal anti-profilin 1 (1:10,000, ab124904, Abcam, Cambridge, UK), mouse monoclonal anti-cofilin (1:1000, ab54532, Abcam, Cambridge, UK) and rabbit polyclonal anti-annexin A2 (1: 1000, ab41803, Abcam, Cambridge, UK).

## Figures and Tables

**Figure 1 ijms-23-02572-f001:**
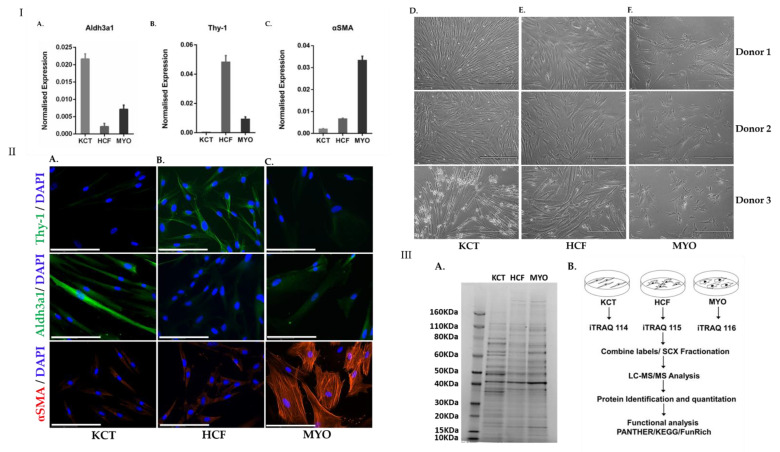
Corneal stromal cell differentiation and cell-specific gene expression markers. (**I**) (A) Aldh3a1 gene specific for keratocytes was observed to have higher expression in KCT, confirming the phenotype. (B) Genetic marker Thy-1 for fibroblasts showed higher expression levels in HCF compared to keratocytes and myofibroblasts. (C) Expression of α-SMA, a characteristic marker for myofibroblasts, was observed to be higher in myofibroblasts, validating the morphological changes. Data represent results from three technical replicates. (D) Corneal keratocytes (KCT) with a distinct dendritic morphology upon culture in serum-free DMEM–F12 medium (96 h) as shown in three donors. (E) Corneal fibroblasts (HCF) showing proliferation and confluency (36 h) in DMEM-F12 with 10% FBS as shown in all four donors. (F) Corneal myofibroblast (MYO) differentiated from HCF (120 h) upon treatment with TGF-β (1 ng/mL) as shown in all four donors. Scale bar: 100 µm. (**II**) Immunocytochemistry of primary keratocytes, fibroblasts and myofibroblasts. (A) Thy-1, Aldh3A1 and alpha SMA expression as shown in keratocytes (KCT). (B) Thy-1, Aldh3A1 and alpha SMA expression as shown in Human Corneal Fibroblasts (HCF). (C) Thy-1, Aldh3A1 and alpha SMA expression as shown in Corneal myofibroblast (MYO). In the top lane, Thy 1 expression (green fluorescence) was higher in fibroblasts as compared to keratocytes and myofibroblasts. In the middle lane, Aldh3A1 marker (green fluorescence) was expressed higher in keratocytes as compared to fibroblasts and myofibroblasts. In the bottom lane, the expression of alpha SMA marker (red fluorescence) expression was higher in myofibroblasts. Nucleus was stained with fluorescent DAPI (blue). These data confirmed the differentiation of fibroblasts into keratocytes and myofibroblasts with expression of specific markers. All the images were taken at 40× magnification (scale bar: 75 µm). (**III**) (A) SDS-PAGE of differentiated corneal stromal cells. Equal concentrations (20 µg) of whole cell lysate from each differentiated cell type—keratocytes (KCT), human corneal fibroblasts (HCF) and myofibroblasts (MYO)—revealed distinct protein profiles. (B) Quantitative proteomics workflow. A 3-plex iTRAQ experiment was performed to study the alterations in the protein expression of the differentiated corneal stromal cells. This experiment was performed in four biological replicates.

**Figure 2 ijms-23-02572-f002:**
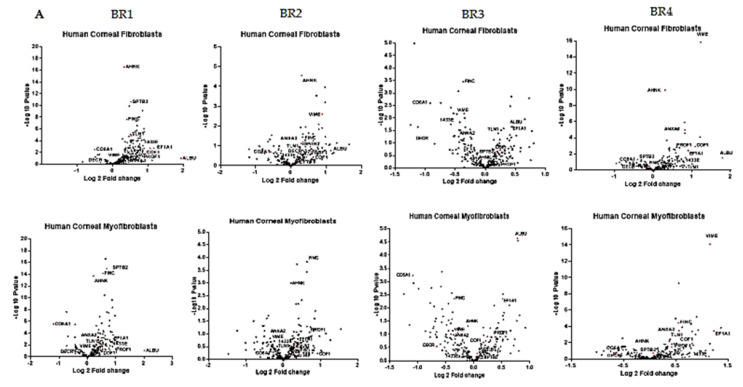
(**A**) Volcano plot analyses for differentially expressed proteins in fibroblasts and myofibroblasts across all four biological replicates (BR1–BR4). A few important proteins are highlighted in red and labeled in all volcano plots. The volcano plot depicts all statistically significant 269 differentially expressed proteins. The fold-change of differentially expressed proteins on the x-axis vs. the statistical significance (−log *p*-value) on the y-axis is shown. (**B**) Hierarchical clustering analysis of 105 proteins (protein coverage 30%) identified in fibroblasts and myofibroblasts in all four biological replicates. The red color represents high gene expression, and blue represents low protein expression. (**C**) The biological classification of 270 proteins into different protein classes is shown as a pie chart with nucleic acid binding and cytoskeletal protein amongst the top ones.

**Figure 3 ijms-23-02572-f003:**
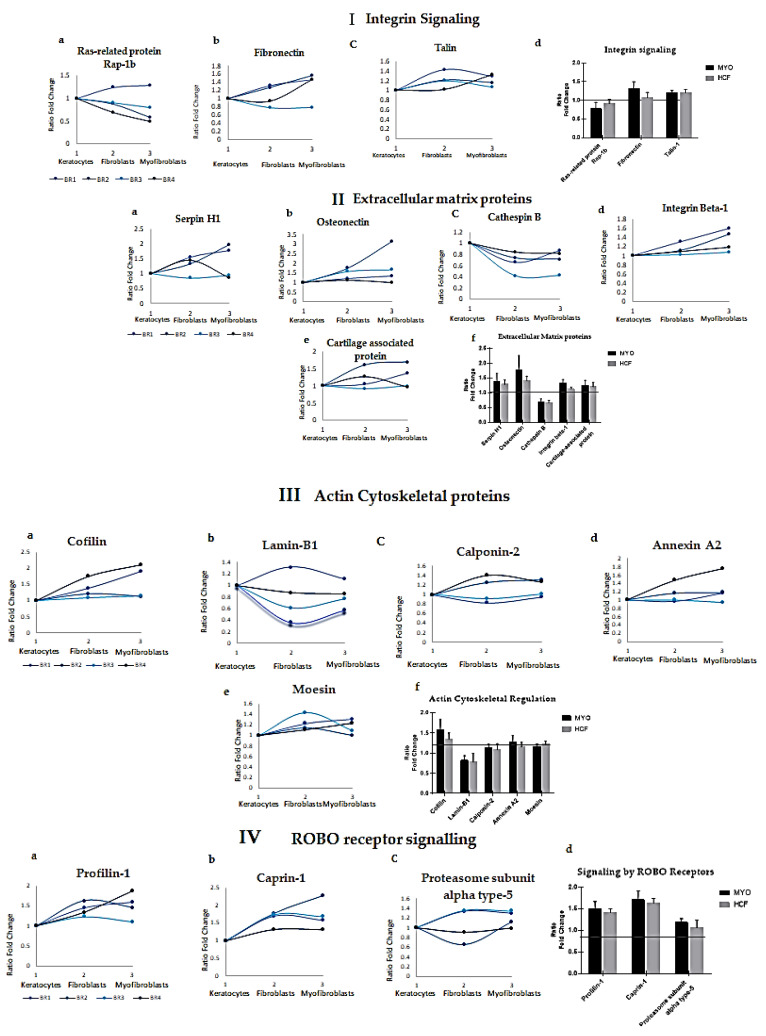
Significantly altered functional classes of proteins during stromal cell differentiation. Panther, KEGG and Reactome pathway analysis of the significantly altered proteins revealed actin cytoskeleton regulation, integrin signaling pathway, ROBO receptor signaling proteins and extracellular matrix proteins as the most significantly altered pathways in corneal stromal cell differentiation. (**I**) Integrin signaling. (a–c) represent the expression of Rap-1b, F and TLN in fibroblasts and myofibroblasts across all biological replicates. (d) Data represented as standard deviation from all four biological replicates, Ras-related protein (HCF (FC = 0.91), MYO (FC = 0.77)), fibronectin (HCF (FC = 1.07), MYO (FC = 1.31)), talin (HCF (FC = 1.20) and MYO (FC = 1.20)). (**II**) Extracellular matrix proteoglycan proteins. (a–e) represent the expression of SERPINH1, OSN, CTSB, ITGB1 and CRTAP in fibroblasts and myofibroblasts across all biological replicates. (f) Data represented as standard deviation from all four biological replicates, serpin H1 (HCF (FC = 1.28), MYO (FC = 1.38)), osteonectin (HCF (FC = 1.41), MYO (FC = 1.78)), cathepsin B (HCF (FC = 0.66), MYO (FC = 0.70)), integrin beta-1 (HCF (FC = 1.13), MYO (FC = 1.33)) and cartilage-associated protein (HCF (FC = 1.21), MYO (FC = 1.25)). (**III**) Actin cytoskeletal-related proteins. (a–e) represent the expression of COF, LAMB1, CNN2, ANXA2 and MSN in fibroblasts and myofibroblasts across all biological replicates. (f) Data represented as standard deviation from all four biological replicates, cofilin (HCF (FC = 1.35), MYO (FC = 1.57)), lamin B1 (HCF (FC = 0.82), MYO (FC = 0.79)), calponin-2 (HCF (FC = 1.09), MYO (FC = 1.13)), annexin A2 (HCF (FC = 1.15), MYO (FC = 1.26)) and moesin (HCF (FC = 1.22), MYO (FC = 1.15)). (**IV**) ROBO receptor signaling proteins. (a–c) represent the expression of PFN1, CPN1 and PMSA1 in fibroblasts and myofibroblasts across all biological replicates. (d) Data represented as standard deviation from all four biological replicates, profilin 1 (HCF (FC = 1.41), MYO (FC = 1.50)), caprin-1 (HCF (FC = 1.63), MYO (FC = 1.70)) and proteasome subunit alpha type 1 (HCF (FC = 1.06), MYO (FC = 1.18)).

**Figure 4 ijms-23-02572-f004:**
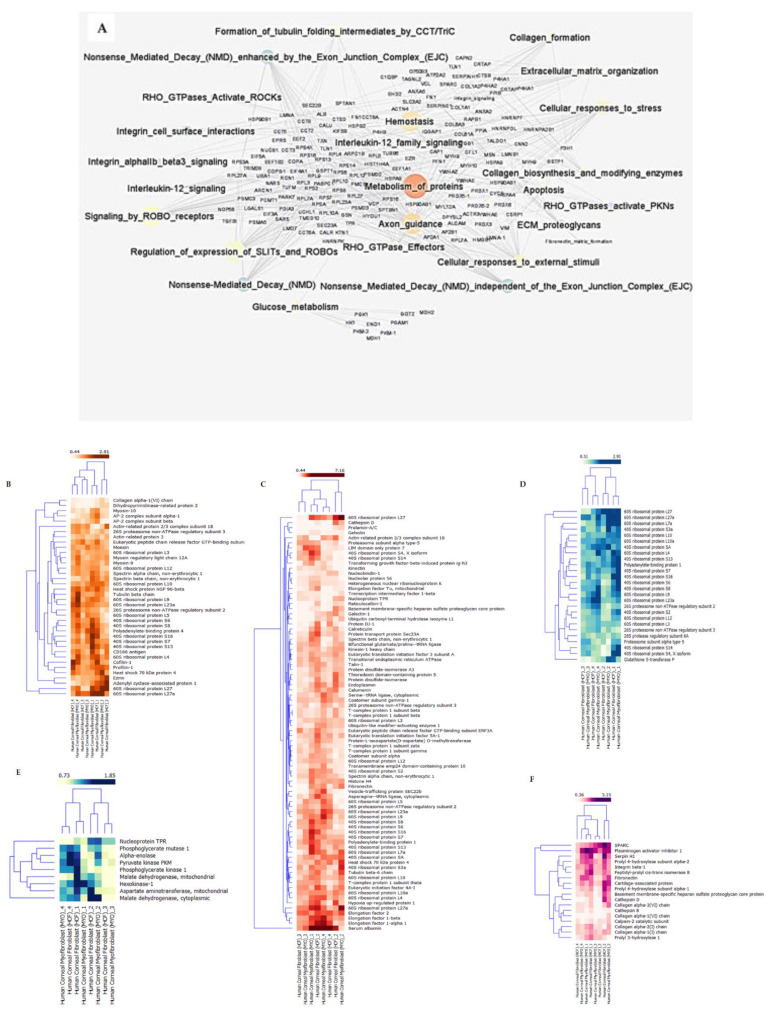
Edge-betweenness-centrality-based network visualization of protein and pathway interactions and hierarchical clustering of highest-degree pathways. (**A**) Interactome map illustrating interaction of important pathways with the proteins based on the edge-betweenness parameters. Metabolism of proteins showed high edge-betweenness with TPR, P4B, FN1, HSPG2; axon guidance interactions showed the lowest number of shortest paths interacting primarily with MSN protein in a network, whereas glucose metabolism interactions showed the maximum number of shortest paths in a network. Regulation of expression of SLITs and ROBOs showed high edge-betweenness in the network. (**B**) Hierarchical clustering of highest-degree pathway axon guidance. A total of 38 proteins were identified to be involved in axon guidance pathways. Data represent an expression pattern in all four biological replicates. (**C**) Hierarchical clustering of highest-degree pathway metabolism of proteins. A total of 81 proteins were identified to be involved in metabolism of protein pathway. Data represent expression pattern in all four biological replicates. (**D**) Hierarchical clustering of highest-degree pathway SLIT-ROBO pathway. A total of 26 proteins were identified to be involved in SLIT-ROBO pathway. Data represent expression pattern in all four biological replicates. (**E**) Hierarchical clustering of highest-degree pathway glucose metabolism pathway. A total of 9 proteins were identified to be involved in glucose metabolism. Data represent expression pattern in all four biological replicates. (**F**) Hierarchical clustering of highest-degree pathway extracellular matrix pathways. A total of 18 proteins were identified to be involved in extracellular matrix. Data represent expression pattern in all four biological replicates.

**Figure 5 ijms-23-02572-f005:**
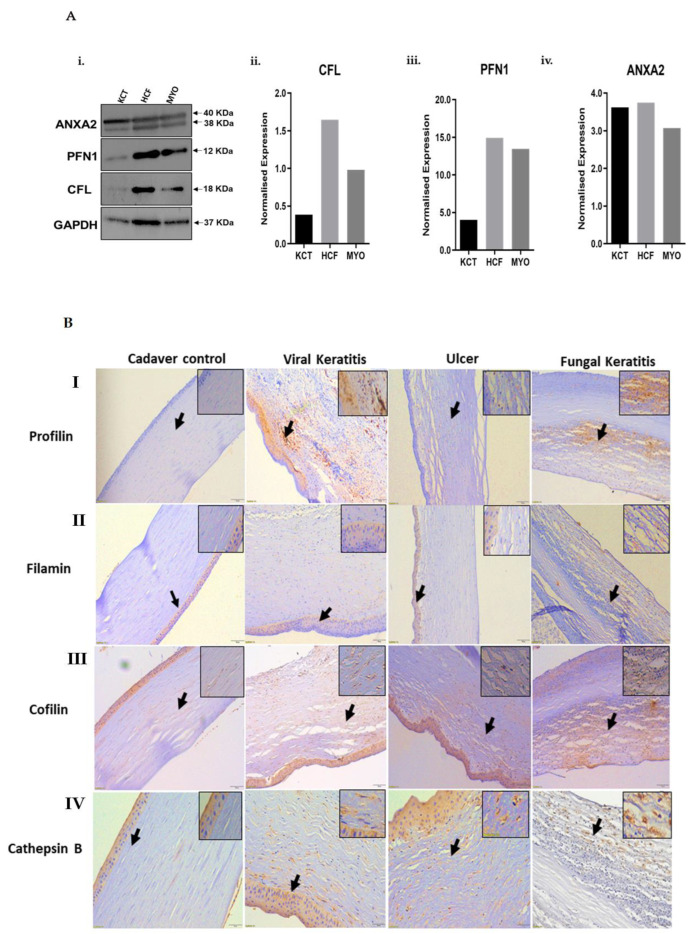
(**A**) (i) Western blotting: Ten micrograms of total cellular proteins from various cell lines were electrophoresed through a denaturing polyacrylamide gel, electroblotted and hybridized with actin cytoskeletal proteins cofilin (COF), profilin 1 (PROF1) and annexin A2 (ANXA2) in keratocytes, fibroblasts and myofibroblasts as normalized to glyceraldehyde-3-phosphate dehydrogenase (GAPDH). (ii–iv) Densitometry analysis of proteins: Quantification levels of cofilin, profilin 1 and annexin A2 in KCT, HCF and MYO cell lines. All proteins have been analyzed on the same gel. (**B**) (I–IV) Immunohistochemical analysis of profilin 1, cofilin, filamin A and cathepsin B. Data are from a representative experiment including the fibrotic and non-fibrotic areas of patient cornea and a control cornea. (I) Profilin 1 was predominantly expressed in stroma of viral keratitis and fungal keratitis group and to some extent in stroma of ulcer group. Higher profilin 1 expression in the fibrotic stroma is the representative myofibroblastic activation of stromal fibrocytes as shown in the top lane. Control group did not show any expression of profilin 1. (II). Filamin A expression was significantly higher in stroma of fungal keratitis group as compared to the control. Higher expression of filamin A in stromal scarred tissue indicates the presence of myofibroblast-mediated stromal fibrosis. (III) Cofilin showed higher expression in fibrotic stroma of all groups (viral keratitis, ulcer and fungal keratitis) as compared to the control. Higher expression of cofilin indicates presence of mature myofibroblasts in the scarred corneal stroma. (IV). Cathepsin B expression was significantly higher in stroma of fungal keratitis, viral keratitis and ulcer as compared to the control. Higher cathepsin B expression indicates myofibroblastic activation of stromal fibrocytes. The arrow represents the scarred area in the tissue as shown by positive staining.

**Figure 6 ijms-23-02572-f006:**
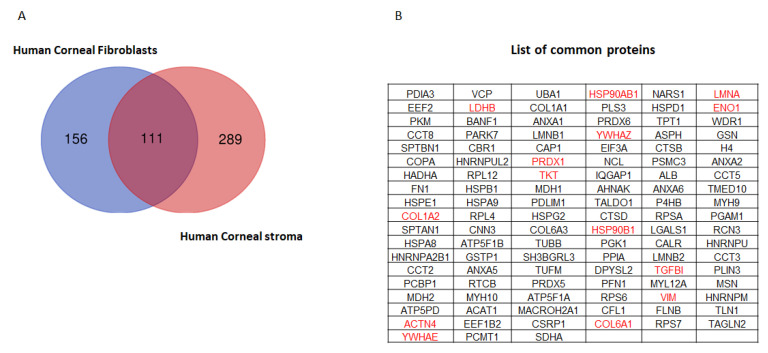
(**A**) Venn diagram of common and unique proteins compared between human corneal fibroblasts and human corneal stroma. A total of 111 proteins were found to be common between human corneal fibroblasts and human corneal stroma (cadaver). (**B**) List of common proteins between human corneal fibroblasts and human corneal stroma. The proteins highlighted in red are known to play a significant role in early wound healing process.

**Figure 7 ijms-23-02572-f007:**
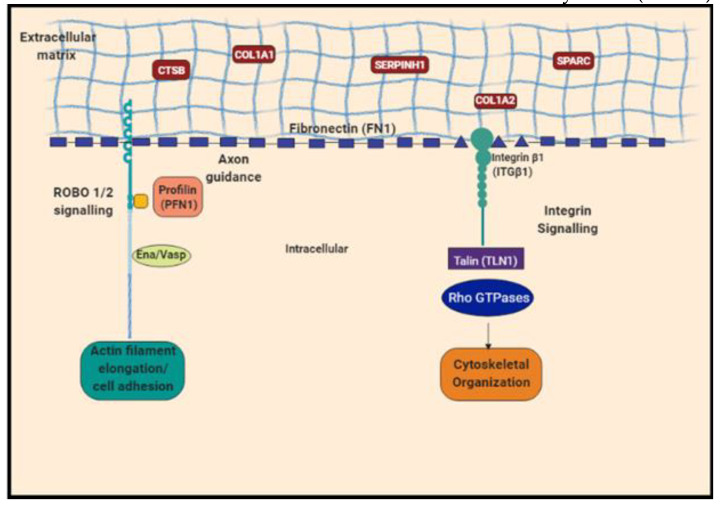
Schematic representation of key proteins involved in axon guidance, ROBO signaling and integrin signaling regulating the actin filament elongation/cell adhesion and cytoskeletal organization.

**Table 1 ijms-23-02572-t001:** Reactome analysis depicting the significantly altered pathways during corneal stromal cell differentiation.

Pathway Identifier	Pathway Name	Proteins Identified	Total Proteins	Entities *p* Value	Entities FDR
R-HSA-422475	Axon guidance	52	557	1.11 × 10^−16^	2.95 × 10^−^^14^
R-HSA-975956	Nonsense-mediated decay (NMD) independent of the exon junction complex (EJC)	23	96	6.66 × 10^−^^16^	1.06 × 10^−^^13^
R-HSA-9010553	Regulation of expression of SLITs and ROBOs	28	172	6.11 × 10^−^^15^	5.36 × 10^−^^13^
R-HSA-975957	Nonsense-mediated decay (NMD) enhanced by the exon junction complex (EJC)	23	117	3.94 × 10^−^^14^	2.09 × 10^−^^12^
R-HSA-927802	Nonsense-mediated decay (NMD)	23	117	3.94 × 10^−14^	2.09 × 10^−12^
R-HSA-376176	Signaling by ROBO receptors	30	218	4.71 × 10^−14^	2.17 × 10^−12^
R-HSA-9020591	Interleukin-12 signaling	13	46	2.09 × 10^−10^	6.08 × 10^−9^
R-HSA-447115	Interleukin-12 family signaling	13	56	2.18 × 10^−9^	5.66 × 10^−8^
R-HSA-392499	Metabolism of proteins	85	2010	1.14 × 10^−7^	2.29 × 10^−6^
R-HSA-1650814	Collagen biosynthesis and modifying enzymes	11	67	1.13 × 10^−^^6^	2.03 × 10^−^^5^
R-HSA-1474290	Collagen formation	12	90	3.09 × 10^−^^6^	4.95 × 10^−^^5^
R-HSA-389960	Formation of tubulin folding intermediates by CCT/TriC	7	26	4.54 × 10^−^^6^	7.27 × 10^−^^5^
R-HSA-2262752	Cellular responses to stress	26	408	1.08 × 10^−^^5^	1.62 × 10^−^^4^
R-HSA-3000178	ECM proteoglycans	9	76	1.29 × 10^−^^4^	0.00168208
R-HSA-8953897	Cellular responses to external stimuli	27	505	1.42 × 10^−^^4^	0.001704805
R-HSA-109581	Apoptosis	14	178	1.56 × 10^−^^4^	0.001872861
R-HSA-70326	Glucose metabolism	10	98	1.83 × 10^−^^4^	0.002199433
R-HSA-1474244	Extracellular matrix organization	19	301	1.93 × 10^−^^4^	0.002310664
R-HSA-216083	Integrin cell surface interactions	7	85	0.005395875	0.032375248
R-HSA-195258	RHO GTPase effectors	15	295	0.006736196	0.034468636
R-HSA-1566977	Fibronectin matrix formation	2	6	0.009769066	0.047246897
R-HSA-109582	Hemostasis	28	722	0.011423688	0.047246897
R-HSA-5627117	RHO GTPases activate ROCKs	3	19	0.011811724	0.047246897
R-HSA-5625740	RHO GTPases activate PKNs	5	63	0.020332384	0.064732946
R-HSA-354192	Integrin alpha IIb beta3 signaling	3	28	0.032248972	0.08304909
R-HSA-9006921	Integrin signaling	3	28	0.032248972	0.08304909

## Data Availability

All pertinent data is presented in the manuscript and associated Appendix A. Raw spectral data can be obtained from the corresponding authors.

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
