# Peer review of "Quantitative Proteomics Reveals Molecular Network Driving Stromal Cell Differentiation: Implications for Corneal Wound Healing"

_ijms, 2022, doi:10.3390/ijms23052572_

Round 1

Reviewer 1 Report

The manuscript presented interesting novel data on a significant scientific subject, however it can be improved noticeably, as per the few following recommendations:

The amount of the cited published literature seems to be insufficient for adequate demonstration of the previous relevant work both in the Introduction and Discussion sections. 

When describing the stromal cell proteins as revealed by 1-D PAGE (which is quite an obsolete method of protein analysis) the authors indicated (lines 333-334) that similar protein pattern was found in earlier work (ref.21) referring to a Coomassie blue-stained gel which in fact did not look quite similar to their SDS-gel (Fig.1, III). Moreover, the data on more than 250 proteins identified in that study (ref.21) were obtained using 2-D PAGE which is a rather high-resolution method yielding significant amount of data suitable to be discussed with respect to the interesting data presented in this manuscript.

The manuscript requires significant improvement in writing, including extensive editing by native speaker, to correct inaccuracies, such as “immune staining” in Fig.5 legend, “incubated with primary antibody Filamin A, Profilin” (l. 479) or multiple mistakes/typos (such as l.155-156 and on l.173) that makes quite hard to comprehend the authors’ writing.

Also, it may be appropriate if the list of the publications has been made in a conventional manner, i.e., in accord with the order of their citation in the text (e.g., ref.6 is followed by ref.13-15, and then by ref.11).

Author Response

Response to Reviewer’s Comments

We thank the reviewer for reviewing the manuscript and for very insightfully highlighting the important points in our manuscript.

Point 1: The amount of the cited published literature seems to be insufficient for adequate demonstration of the previous relevant work both in the Introduction and Discussion sections.

Response to Point 1: We agree with the reviewer and have now added more references with additional discussion and context of relevant prior literature in Introduction and Discussion sections. Kindly refer to the Introduction and Discussion sections to evaluate the changes.

Point 2: When describing the stromal cell proteins as revealed by 1-D PAGE (which is quite an obsolete method of protein analysis) the authors indicated (lines 333-334) that similar protein pattern was found in earlier work (ref.21) referring to a Coomassie blue-stained gel which in fact did not look quite similar to their SDS-gel (Fig.1, III). Moreover, the data on more than 250 proteins identified in that study (ref.21) were obtained using 2-D PAGE which is a rather high-resolution method yielding significant amount of data suitable to be discussed with respect to the interesting data presented in this manuscript.

Response to Point 2: We thank the reviewers for this observation. Karring et al showed 1D SDS gel of soluble fraction from human corneal fibroblasts. We observed 3 intense bands between 45 and 60 kDa and few less intense bands above 60 kDa in human corneal fibroblast which is similar to that shown by Karring et al in 1-D PAGE. Many less intense low molecular weight bands (< 30kDa) were also observed in our data which was also in accordance with Karring et al 1-D PAGE. Please note that the intensity of the staining is different between the Karring et al image and ours. In order to clarify the patterns, we have now added molecular weight marker sizes to the gel image. Moreover, our experiment highlights the protein profile differences by running 1-D PAGE between the Human Corneal Keratocytes, Human Corneal Fibroblasts and Human Corneal Myofibroblast. We agree that 1-D gel is a rather low resolution method, however, through this image we simply sought to give the reader a glimpse of the slight changes in patterns of the whole proteome in a simple visual manner. For higher resolution understanding of the differences between the cell types, the LC-MS was performed.

We agree with the reviewer that the data on more than 250 proteins identified in the study (ref.21) should be discussed with respect to the data presented in this manuscript. We have discussed the results with reference to data reported by Karring et al in the revised discussion.

Point 3: The manuscript requires significant improvement in writing, including extensive editing by native speaker, to correct inaccuracies, such as “immune staining” in Fig.5 legend, “incubated with primary antibody Filamin A, Profilin” (l. 479) or multiple mistakes/typos (such as l.155-156 and on l.173) that makes quite hard to comprehend the authors’ writing.

Response to Point 3: Considering reviewer’s suggestion, the manuscript was extensively edited by an expert. We have made corrections in Figure 5 legend, and also made necessary changes to Method section and corrected typographical errors throughout the manuscript. We hope that the changes made will improve the clarity of the manuscript and discussion.

Point 4 Also, it may be appropriate if the list of the publications has been made in a conventional manner, i.e., in accord with the order of their citation in the text (e.g., ref.6 is followed by ref.13-15, and then by ref.11).

Response to Point 4:  We note this observation. We have made the necessary corrections throughout the text and now the references are in accord with the order of their citation in the text.

Reviewer 2 Report

The authors examined by proteomic analysis the differentiation of corneal stromal keratocytes to fibroblasts and myofibroblasts that occur during corneal wound healing. Cultured human corneal fibroblasts were back-differentiated into keratocytes in serum free media or into myofibroblasts by TGF-β. Proteins in cell lysates from different donors were trypsin-digested, labeled using a 3-plex iTRAQ kit, and subjected to mass spec analysis. The functional analysis revealed an enriched set of proteins involved in differentiation of human corneal stromal cells. The selected proteins were validated by immunohistochemistry. It is shown that proteins involved in integrin signaling (Ras-RAP1b, TLN and FN), ROBO-Slit pathways (PFN1, 30 CAPR1, PSMA5), extracellular matrix proteins (SERPINH1, SPARC, ITGβ1, CRTAP) had increased expression in corneal fibroblasts and myofibroblasts compared to keratocytes indicating regulatory roles in wound healing. It is concluded that corneal keratocyte differentiation is associated with activation of molecular pathways in the repair phenotypes exemplified by fibroblasts and myofibroblasts. Cytoskeletal profilin and talin could be important for stromal healing and may have potential as molecular targets for treating corneal fibrosis.

The paper presents the first account of global proteomics in three populations of corneal stromal cells and thus is very important for the corneal field.

The authors are invited to address the following minor concerns:

  1. Figure 1(II). Please add “DAPI” to the antigen description at the left of the pictures, as well as mention it in the legend. Otherwise, it is difficult to appreciate that in keratocytes, Aldh3a1 is in the cytoplasm, but in other cell types only the nuclei are DAPI-positive.

Additionally, the SMA staining is not convincing even at high magnification. Please check the pictures and provide better ones. Magnification bars should be included. Also, the gel in III(A) is too dark and lacks contrast. Please increase contrast and brightness, which is scientifically acceptable.

  1. The section “Determination of terminal differentiation of corneal stromal cells” should probably be renamed for clarity as “Determination of differentiation state of corneal stromal cells”.
  2. In Figure 5, all three westerns have the same housekeeper lanes. Please state that all proteins have been analyzed on the same gel; then, the same housekeeper panel is legit. If the gels were different, please change the housekeeper panels to reflect separate gels. Please add designation “A” to the upper panel.

In panel B, please explain what kinds of cells are expected to be accumulated in each specific disease for easier understanding. As is, the legend is not informative. The same concern applies to the text.

  1. I may have missed it, but please indicate how many samples were analyzed and whether there was any pooling of the samples from different cases. It would be helpful to add some details about the fibroblast cell line (how it was obtained, whether it is transformed/tumorigenic or not).
  2. Statistical significance is usually set as <0.05, not as ≤ 0.05.
  3. Please add catalog numbers of antibodies.

Reviewer 3 Report

Nishtala et. al. have done a comprehensive study of changes in gene profiles of corneal stromal cells and their 3 major phenotypes. They show some interesting molecules that may play a role in wound healing, but they do not show any definitive data as to their function. Some of the figures are not convincing, and require revision.

Comments:

1. (Figure 1): The keratocytes in Fig. 1 (II) are spindle shaped and do not look at all like the typical morphology of keratocytes shown in Fig. 1 (I. D). How do they describe this discrepancy? They may need to redo the immunohistochemistry with cells that show the keratocyte morphology.

2. (Figure 1): The background fluorescence for alpha SMA (Fig. 1 II) is very different between the groups. Please present figures with the same exposure levels.

3. (Figure 2 and 3): The Figures seem to overlap on PDF. Cannot see the Hierarchical clustering analysis in Fig. 2

Minor comments:

  1. (Line 52) Brackets are missing for reference no. 4

Author Response

Response to Reviewer’s Comments:

Point 1 (Figure 1): The keratocytes in Fig. 1 (II) are spindle shaped and do not look at all like the typical morphology of keratocytes shown in Fig. 1 (I. D). How do they describe this discrepancy? They may need to redo the immunohistochemistry with cells that show the keratocyte morphology.

Response 1: We agree with the reviewer’s observation. As fibroblasts were obtained from four different donors and then differentiated into keratocytes, we observed slight changes in the morphology of differentiated keratocytes obtained from different donors. Fig.  1 (II) is the representative image of the morphology of keratocytes, fibroblasts and myofibroblasts. To avoid discrepancy, we have now presented the microscopic images from three donors in Figure 1 (II). The Immunohistochemistry images are from one of the donors each as a representative image. Now we have shown enlarged view of the highlighted area which shows clear myofibroblast phenotype (Figure 1 (II) alpha sma Bottom panel). We have considered the changes to improve the clarity of the image.

Point 2. (Figure 1): The background fluorescence for alpha SMA (Fig. 1 II) is very different between the groups. Please present figures with the same exposure levels.

Response 2: We agree with the reviewer that the background fluorescence was quite high for alpha SMA image. All the images presented in (Figure 1 II) have same exposure levels. We agree that the background fluorescence for alpha SMA was very different as it was not corrected. We have rectified the error. Please refer revised (Fig. 1 II)

Point 3 (Figure 2 and 3): The Figures seem to overlap on PDF. Cannot see the Hierarchical clustering analysis in Fig. 2

Response 3 We are sorry for the inconvenience. We will upload the edited PDF version of the manuscript and will make sure that it doesn’t cause any overlaps after conversion.

Minor comments:

Point 1. (Line 52) Brackets are missing for reference no. 4

Response 1. We have added the brackets to reference no. 4 in the manuscript.

Round 2

Reviewer 3 Report

No further comments

Author Response

There were no further comments by the reviewer.